# Dynamic and Active THz Graphene Metamaterial Devices

**DOI:** 10.3390/nano12122097

**Published:** 2022-06-17

**Authors:** Lan Wang, Ning An, Xusheng He, Xinfeng Zhang, Ao Zhu, Baicheng Yao, Yaxin Zhang

**Affiliations:** 1Yangtze Delta Region Institute (Huzhou), University of Electronic Science and Technology of China, Huzhou 313001, China; wanglan@uestc.edu.cn; 2Key Laboratory of Optical Fiber Sensing and Communications (Education Ministry of China), University of Electronic Science and Technology of China, Chengdu 610054, China; anning_ph.d@std.uestc.edu.cn; 3School of Electronic Science and Engineering, University of Electronic Science and Technology of China, Chengdu 610054, China; hexs159@163.com (X.H.); 202121020909@std.uestc.edu.cn (X.Z.); 202152022117@std.uestc.edu.cn (A.Z.)

**Keywords:** terahertz wave, graphene, metamaterial

## Abstract

In recent years, terahertz waves have attracted significant attention for their promising applications. Due to a broadband optical response, an ultra-fast relaxation time, a high nonlinear coefficient of graphene, and the flexible and controllable physical characteristics of its meta-structure, graphene metamaterial has been widely explored in interdisciplinary frontier research, especially in the technologically important terahertz (THz) frequency range. Here, graphene’s linear and nonlinear properties and typical applications of graphene metamaterial are reviewed. Specifically, the discussion focuses on applications in optically and electrically actuated terahertz amplitude, phase, and harmonic generation. The review concludes with a brief examination of potential prospects and trends in graphene metamaterial.

## 1. Introduction

Terahertz radiation (0.1 THz–10 THz) is situated between microwave and infrared light, possessing the properties of photonics and electronics [1]. The past 20 years have seen a tremendous growth in terahertz science and technology, including diverse research on high-rate wireless communications, high-resolution environment sensing, and imaging research for medical applications [2,3,4,5,6]. Terahertz technology requires significant advances in the overlapping areas of materials and device design, making terahertz components for efficient operation, detection, and generation compact and economical commercially viable in the future.

A wide variety of active materials have been used for terahertz applications, including traditional bulk semiconductor materials, two-dimensional materials, phase-change materials, and liquid crystal materials, which are easily modulated when the carrier concentration is under externally applied electrical or optical stimulus. Graphene stands out due to its easy integration and its unique properties over a broad spectrum, especially in the terahertz band. [7,8,9,10]. It introduces a universal absorption for visible and near-infrared frequencies due to the generation of electron–hole pairs caused by interband transitions [11,12]. A Drude-like conductivity is observed at far-infrared and terahertz frequencies, corresponding to the absorption of intraband carriers [13,14]. Furthermore, graphene shows an exceptional nonlinear optical response due to the absence of bandgap and linear energy–momentum dispersion [15,16]. The graphene nonlinearities have been demonstrated at infrared and visible frequencies, originating from interband electron dynamics, including saturable absorption [17], higher-harmonic generation [18,19,20,21,22], and four-wave mixing [23,24]. Nevertheless, in the technologically important terahertz region, the up to seventh-harmonic generation was confirmed experimentally until 2018 [25]. The strong intraband nonlinearity of graphene is related to the collective thermal response of its background Dirac electrons under terahertz fields, and the controllable saturable absorption effect and third-harmonic generation have been demonstrated successfully in doped graphene [26,27]. Exploring the THz linear and nonlinear properties of graphene is impressive for developing high-speed optoelectronic devices such as graphene-based transistors, photodetectors, and laser mode-locking.

Metamaterials is a rapidly developing direction that has been extensively boosted to metasurfaces and meta-atom structures [28,29,30,31,32]. Actively-tunable metamaterials can significantly reduce the size of optical devices and enhance light-matter interaction by field localization, which contributes to improving the light modulation capability of graphene, limited by the ultra-thin thickness of only 0.34 nm [33]. Graphene metamaterial comprises graphene with conventional artificial metamaterial structures or alternatively stacked graphene and dielectric layers [34,35]. Thanks to the carefully-designed geometric shapes, graphene metamaterial offers an ambitious platform for beam transformation, leading to extensive applications, such as modulators, light emitters, detectors, and terahertz lenses. In addition, multilayer graphene hyperbolic metamaterial with strong light absorption is proposed, where graphene-based grating consists of alternating graphene layers separated by lossless dielectric [36]. Furthermore, transmission lines (TL) based on graphene metamaterials have also been extensively studied, which can control the phase constant and the characteristic impedance of the transmission line [37,38]. This further controllability of circuit parameters paves the way for designing on-chip structures with more compact dimensions, a higher performance, and novel functionalities. Through careful design, graphene-based non-linear transmission line structures and mode-selective transmission line structures can be used in various integrated devices, such as modulators, phase shifters, and antennas [39,40,41]. However, limited by the traditional complicated and time-consuming graphene manufacturing process, the manufacturing of multilayer and large-area patterning graphene metamaterials remains a challenge.

This review aims to discuss the development of terahertz metamaterial devices utilizing graphene. Section 2 introduces the linear and the nonlinear properties of graphene in the THz region. Section 3 focuses on various applications of graphene metamaterials. Finally, Section 4 provides conclusions and a future outlook on this field.

## 2. Fundamentals of Graphene

### 2.1. Linear Optical Property

The optical properties of graphene originate from the two-dimensional electronic band where the conduction and valence bands contact at the Dirac point, forming the honeycomb lattice. The optical response of graphene electrons at visible and infrared frequencies is independent of incident light frequency and can be described by universal conductivity $G=e^{2}/4\hbar$. Compared to other semiconductor or metal materials with the same atom layer thickness, monolayer graphene has an extreme optical absorption (with πα = 2.3%) [42]. In contrast, the reflection (<0.1%) is almost negligible in the visible light range. The opacity is determined only by the precision structure constant α = E^2^/ℏc [43]. Without inter-layer electron coupling, the optical absorption of uniform multilayer graphene is dependent on the layers, and the opacity increases linearly by 2.3% with the stacking of layers.

Linear response theory is applied to the light-matter interaction of graphene at far-infrared and terahertz frequencies. Under a random phase approximation condition, the dynamic optical conductivity of graphene is expressed as [44,45]:(1)$$\begin{array}{l} \sigma_{g}\left(\omega ,\mu_{c},\Gamma \left(or\;\tau^{-1}\right),T\right)=\frac{-ie^{2}\left(\omega +i2\Gamma \right)}{\pi \hbar^{2}}\left[\frac{1}{{\left(\omega +i2\Gamma \right)}^{2}}\right.\int_{0}^{\infty }\varepsilon \left(\frac{\partial f_{d}\left(\varepsilon \right)}{\partial \varepsilon }-\frac{\partial f_{d}\left(-\varepsilon \right)}{\partial \varepsilon }\right)d\varepsilon -\\ \;\;\;\;\;\;\;\;\;\;\left.-\int_{0}^{\infty }\frac{f_{d}\left(-\varepsilon \right)-f_{d}\left(\varepsilon \right)}{{\left(\omega +i2\Gamma \right)}^{2}-4{\left(\varepsilon /\hbar \right)}^{2}}d\varepsilon \right] \end{array}$$
where $f_{d}\left(\varepsilon \right)=1/\left[\exp\left(\varepsilon -\mu_{c}/k_{B}T\right)+1\right]$ is the Fermi distribution; Γ is the carrier scattering rate determined by defect scattering, ionized impurity scattering, and acoustic and optical phonon scattering; and *μ_c_* is chemical energy dependent on carrier density. Graphene conductivity comprises intraband and interband conductivity, which are induced by electron–phonon scattering relaxation and interband electron transition, respectively. When $\left|\mu_{c}\right|\gg K_{B}T$, intraband conductivity is expressed by the Drude–Boltzmann model similar to metal, and it is controlled by chemical energy *μ_c_*. For interband conductivity, there is a critical frequency $\omega_{inter}$ related to chemical energy $h\omega_{inter}=\mu_{c}$. The frequency is lower than $\omega_{inter}$, and the interband conductivity is almost negligible in the total conductivity. Reference [46] presents the detailed evolution of graphene optical conductivity from sub-THz to near-IR with frequency, corresponding to different modulation mechanisms. In the sub-THz region, graphene is usually treated as a frequency-independent film, and the terahertz wave is controlled by the Fermi level. Surface plasmon polarization can be supported on graphene with more manipulation potential in the terahertz and mid-infrared frequency bands.

### 2.2. Nonlinear Optical Property

Due to its gapless band structure and linear energy–momentum dispersion for its electrons, graphene has a robust nonlinear response [8,10,47]. The incident field can induce intraband transition or interband transition of carriers in graphene with the relationship between the Fermi level and the energy of incident photons. The THz frequency is much lower, making the interband transition happen at a low temperature and with undoped graphene (*E_F_* ≈ 0) [15]. Thus, compared with near-infrared and visible light, the incident THz field could generally induce the intraband transition of graphene carriers, as shown in Figure 1a,b. Additionally, for the THz field from free space into the substrate with graphene, the THz field transmission could be described by the Tinkham equation [25]:(2)$${\tilde{E}}_{t}(w)=\frac{2}{1+n_{s}+Z_{0}\tilde{\sigma }(w)}{\tilde{E}}_{in}(w)$$
where ${\tilde{E}}_{in}(w)$ is the incident THz field, *n_s_* is the refractive index of the substrate, *Z*_0_ is the free-space impedance, and $\tilde{\sigma }(w)$ is the intraband conductivity of graphene at THz band. With the incident THz field, the intraband conductivity $\tilde{\sigma }(w)$ is nonlinear, it could be described by solving of the Boltzmann equation as [25,48,49]:(3)$$\tilde{\sigma }\left(w\right)=-\frac{e^{2}v_{F}^{2}}{2}\int_{0}^{\infty }D(E)\frac{\tau (E)}{1-iw\tau (E)}\frac{\partial f_{FD}(E,\mu_{c},T_{e})}{\partial E}dE$$
where *v_F_* = 1 × 10^6^ m s^−1^ is the Fermi velocity of the relativistic Dirac fermions in graphene, τ(E) is the energy-dependent electron scattering time, $D(E)=2\left|E\right|/\pi (\hbar v_{F}{)}^{2}$ is the density of states, and $f_{FD}(E,\mu_{c},T_{e})={\left\{\exp\left[\left(E-\mu_{c}\right)/k_{B}T_{e}\right]+1\right\}}^{-1}$ is the Fermi–Dirac distribution function for chemical potential *μ_c_* and electron temperature *T_e_*.

From Equation (3), we could find that the intraband transition conductivity of graphene is determined by *μ_c_*, electron temperature (*T_e_*), and density of states of the graphene energy bands *D*(*E*). As shown in Figure 1c, the energy of the incident THz is transferred to electrons in graphene, leading to an increase in *T_e_*, and the concomitant decrease of σ. It means that the rise of THz frequency and THz intensity could both lead to a decrease in σ (Figure 1d). In the following section, based on the different responses of graphene THz nonlinearity, we briefly introduce the THz nonlinear mechanism by THz nonlinear absorption, THz high harmonic generation, and different frequency generation (DFG).

#### 2.2.1. Nonlinear Absorption

THz nonlinear absorption is widely used in graphene-based THz modulators [35,51]. Compared with the linear absorption, the nonlinear absorption of graphene THz is mainly reflected in the nonlinear enhancement of the transmittance of THz with the increase of the incident THz intensity or optical pump, as shown in Figure 2a. Such nonlinear enhancement is mainly due to the saturable absorption effect associated with the reduced intraband conductivity, as shown in the following equation [52]:(4)$$T\left(w\right)={1/\left|1+Z_{0}\sigma \left(w\right)/\left(n_{s}+1\right)\right|}^{2}$$
where *Z*_0_ is the vacuum impedance and *n_s_* is the substrate refractive index; *σ*(*w*) is the intraband conductivity of graphene. From Equation (4), we could find that the transmission of the incident THz field depends on the intraband conductivity. When the THz field is incident on the substrate, the absorption of the THz field by graphene’s free carriers results in a transient increase of the electron temperature. Furthermore, according to Equation (3), an elevated electron temperature could decrease the intraband conductivity of graphene [50], which makes the absorption of the THz field decrease.

#### 2.2.2. High Harmonic Generation

As mentioned above, the strong THz field incident on graphene increases the carriers’ temperature, which causes a decrease in the graphene conductivity. Graphene’s carriers are restored to their original state by phonon emission. In the process, ultrafast carriers heating and cooling lead to nonlinear temporal modulation of THz absorption and make the instantaneous graphene conductivity modulate nonlinearly. The nonlinear conductivity leads to the generation of the nonlinear current of graphene (*j*(*t*) = *σE*(*t*)) [25]. If the frequency of the incident THz is *f_0_*, the generated nonlinear current will contain high harmonics for the driving frequency. Due to the centrosymmetric graphene, even-order harmonic generation (2*f*_0_, 4*f*_0_ …) is forbidden, and the generated current contains contributions of odd-order harmonics (*f_0_*, 3*f*_0_, 5*f*_0_ …), which leads to the electromagnetic reemission of high harmonics, as shown in Figure 2b. Unlike the high harmonic generation of the optical band, the high harmonic generation of the THz band relies on the nonlinear current driven by the nonlinear conductivity, which is based on the intraband transition.

#### 2.2.3. Difference Frequency Generation

Due to its isotropic nature, the second-order nonlinearity of graphene could be negligible. However, the second-order polarizability cannot be ignored when considering spatial dispersion [54,55]. For graphene DFG-based THz generation, an increase is provided by graphene’s effective second-order nonlinear coefficients. The process can be described as follows: the electrons in the valence band of graphene absorb a photon and cross the Dirac point to the conduction band, the electrons in the conduction band are then unstable, relax downward, and generate a low-frequency photon, as shown in Figure 2c [53]. Unlike the two nonlinear processes mentioned above, THz generation based on graphene DFG occurs when the incident optical pump and signal satisfy the energy conservation and momentum conservation:(5)$$\begin{array}{l} \hbar f_{sp}+\hbar f_{s}=\hbar f_{p}\\ {\vec{k}}_{sp}+{\vec{k}}_{s}={\vec{k}}_{p} \end{array}$$
where *f_p_* represents the pump light, *f_s_* represents the signal light, and *f_p_* represents the idler light, *ℏ* is the reduced Planck’s constant.

## 3. Applications of Graphene Metamaterial

### 3.1. THz Amplitude Modulation

Graphene is usually used as the active region in the terahertz amplitude modulation metamaterial. The voltage or laser beam controls the Fermi level of graphene; thus, the propagating terahertz wave can be modulated [56,57]. The optical response of graphene metamaterial to an incident THz wave can be characterized by changing the real and the imaginary components of the equivalent complex refractive index, which external excitations can modify. Table 1 shows the recent evolution of graphene metamaterials in amplitude modulation. The modulation depth of graphene modulators has significantly improved, approaching 100%. At the same time, its modulation rate is also increased from kHz to GHz.

According to the applied external fields, the typical approaches are categorized into electro-optic modulation and all-optical modulation. The electronically controlled graphene modulator controls transmission intensity through the applied voltage, with integration, reversibility, and flexibility advantages. Graphene, with unique properties such as its tunable carrier density and high room temperature carrier mobility, is an ideal choice for realizing tunable metamaterial devices. The generation of tightly confined surface plasmons (SPs) in graphene upon excitation by incident light has been demonstrated in various patterns, such as arrayed graphene ribbons (GR), which are also intrinsically tunable metamaterials. By embedding graphene ribbons into the gaps of C-SRR capacitors, Liu et al. achieved near-field coupling between graphene surface plasmons and LC resonances [62]. The LC resonance was localized and enhanced within the capacitive gap, and the local SP resonance was also highly localized near the GR. Therefore, embedding the GR into the C-SRR capacitive gap is an effective way to realize the strong near-field coupling of the two structures, thereby forming a coupled resonant system. The spectral response of the hybrid metamaterial is modulated by tuning the local SP resonance by electrically changing the carrier density. The maximum modulation depth was 60% at the applied voltage of 120 V. Although sufficient modulation was achieved, the required voltage was too high. Wu et al. effectively modulated graphene conductivity at a smaller voltage of 3 V by adding a layer of ionic liquid between the two graphene layers [63], as shown in Figure 3a. The positive and the negative charges accumulate at the interface between graphene and ionic liquid with the voltage applied, attributed to the strong gating effect of the ionic liquid. Therefore, the electric field on graphene is highly enhanced, leading to an efficient tuning of graphene’s Fermi level. The incredibly high carrier concentration leads to a high transmission modulation of up to 99% by stacking two structures.

Compared with ionic liquids, solid-state ionic gels consisting of ionic liquids confined in copolymer blocks are more convenient to modify the carrier distribution of graphene at low voltages. Jung et al. formed an electrical connection between the graphene ribbon and H-shaped units by placing a layer of ionic gel on the metal array [64]. Adjusting the gating voltage from 1 V to −3 V, the maximum change of the transmitted wave reached 49.3%. In addition to amplitude, a maximum phase shift of 68.1° also occurred with the applied voltage. However, it is challenging to achieve amplitude or phase modulation independently. Han et al. proposed the concept of “metamolecules” to overcome the limitation [67]. As shown in Figure 3b, each “metamolecule” contains two independent subwavelength scatterers composed of the metal antenna and the graphene ribbon coupled with each other. The subwavelength noble metals improve radiative coupling to GPRs by reducing the mismatch between free photons and subwavelength scale graphene plasmons. Besides the metal plasmonic structure, metal reflectors are also incorporated to further enhance the light–matter interaction in graphene and electrostatically gate graphene. The general tuning mechanism is based on the modulation of a single resonance in the structure, which limits the degrees of freedom for independent control of amplitude and phase. For this metasurface, the GPRs are isolated from each other, and the graphene Fermi level of each scatterer can be adjusted separately, thereby providing two degrees of freedom to independently modulate the amplitude and the phase of the reflected light. Graphene can also form a tunable modulator with a dielectric structure, not limited to the metal structure [70]. Yao et al. combined graphene with a silicon substrate with rectangular grooves, and the graphene was connected to the electrodes via an ion gel. The dielectric modulator achieved a maximum modulation depth of approximately 70% at a voltage of only 2.5 V. Recently, Zeng et al. presented an all-dielectric graphene metasurface for ultrafast modulation [66]. The thick dielectric between the metal reflector and the resonator array in the modulator caused high voltage bias and limited the modulation speed. To reduce the thickness for a larger capacitance, they replaced part of the dielectric with a material that was conductive to the applied bias but behaved as a dielectric in the mid-infrared range. The proposed high-capacitance structure is as shown in Figure 3c, where the Al_2_O_3_ layer supplies the gate dielectric to improve the capacitance, and the slightly conducting a-Si layer acts as part of the gate electrode. As a result, it significantly reduces the parasitic capacitance with a modulation speed of up to 1 GHz and 90% modulation depth. Furthermore, they applied the modulator to a pixel array and realized a 23 kHz single-pixel imaging.

Besides applying voltage, optical pumping can also effectively change the conductivity of graphene. As shown in Figure 4a, when the pump power increases, the photoinduced conductivity of graphene placed on the SU-8 film decreases, causing the absorption of the incident wave to decrease. This phenomenon is associated with hot carrier generation, an increase in electron temperature, and a significant increase in the scattering rate relative to the carrier concentration in doped graphene [51]. The maximum modulation depth of the reflected wave is 40%, with a carrier attenuation time of 2.7 ps at a pumping flux of 0.7 mJ/cm^2^. Choi et al. combined graphene and nano slot antennas to attain more robust light–graphene interactions at terahertz, enhancing the modulation performance [68]. The modulator has a modulation depth of 80% and an attenuation time of 1.83 ps. The doped graphene is covered on the terahertz nano slot antenna. Without pumping light, the transmission is zero from the enhanced intraband absorption in the graphene due to the strong terahertz field of nano slot antennas. However, once the optical pump is applied to the structure, terahertz transmission increases due to the photo-induced transparency of monolayer graphene. It is demonstrated that the carrier scattering rate will increase under intense pump pulses, enhancing the modulation depth and resulting in the photo-induced transparency. Further, the nanoantenna bandgap can also virtually improve the modulation depth and the attenuation time.

Other approaches, such as localized graphene plasmons and 2D material heterostructures, are also commonly used for amplitude modulators. Localized plasmons can be achieved by patterning graphene, where the light–matter interactions are more notable than unpatterned graphene. Fan et al. pattern monolayer graphene into a periodic array of split-ring resonators as photoexcited graphene modulators, achieving a modulation depth of 92% (in Figure 4b) [65]. The exciting magnetic resonance in graphene SRR is weak due to the high inherent loss, making the plasmon on the surface decay quickly. Photoexcitation is considered a strategy to compensate for the energy dissipation resonantly enhanced in graphene SRR. The optical pump can significantly boost the hardly exciting magnetic resonance in SRR, resulting in transmission decreasing at a resonant frequency from 99% to 8%.

A two-dimensional material heterostructure can also compensate for the defects of traditional materials, exhibiting a strong light–matter interaction. Weis et al. designed an optically controlled modulator based on a graphene–silicon heterostructure [71]. Comparing the number of electron–hole pairs generated under the modulated beam in the graphene and the silicon substrates, graphene was negligible compared to silicon. Free carriers diffused from the silicon substrate into the graphene layer due to the charge gradient until a stable distribution was reached. Carriers diffusing into graphene possess a higher mobility than in silicon, thus resulting in a more pronounced change in conductivity than observed in silicon under the same beam. As discussed above, this results in enhanced modulation of terahertz waves in GOS compared to transmission through pure silicon. A maximum modulation depth of 99% was observed through mutual coupling between the two two-dimensional material layers.

### 3.2. THz Phase Modulation

The graphene and the metamaterial combination present an effective solution to achieving tunable phase shift. Table 2 summarizes the progress of graphene-based terahertz phase modulators. It remains a big challenge to independently control the amplitude and the phase responses and even increase the modulation rate, requiring exploration of new modulation mechanisms and improved material fabrication techniques.

Balci et al. applied graphene capacitors integrated with metallic SRRs, demonstrating a novel electrically tunable phase modulator [72]. The graphene capacitors consisted of two large-area graphene electrodes separated by the dielectric, resulting in efficient mutual gating between the electrodes for high carrier densities. The graphene electrodes introduced additional electrical losses to the SRR due to graphene’s sheet resistance, and it changed the overall impedance required to achieve resonance by capacitive coupling. Based on tunable high mobility carriers on graphene, the resonance of hybrid metamaterials can be tuned efficiently, achieving a phase shift of more than 90° with a gated voltage range from −1.5 V to 1.5 V. The 90° phase modulation range cannot meet the wave front shaping, requiring a more significant phase shift. A multi-layer split ring metasurface composed of graphene and perovskite film was proposed by Yang [69]. Without external stimuli, the carriers in perovskite and graphene are in thermal equilibrium with little effect on higher-order Fano resonances. The optical pump excites the perovskite to generate charge carriers, breaking the thermal equilibrium and correspondingly the photoconductivity of the perovskite increases under pumping. This is also the case with graphene. Therefore, compared with perovskite only, DOM-GPPMM changes the optical conductivity more significantly at the same optical pump. Additionally, the modulation depth is further increased under the external bias voltage. Finally, the resonance can be tuned in the 0.2–1.0 THz band and achieve a maximum modulation depth of 346°, as shown in Figure 5a.

Compared with the phase shift induced by resonance, the Pancharatnam–Berry (PB) phase comes from the rotation angle of the structure. Ding et al. designed a metasurface based on the PB phase to modulate the phase of the cross-polarized transmitted wave [83]. The phase modulation of the structure is related to the polarization state of the transmitted wave. For the transmitted component with cross-polarization, the phase shift is equal to twice the rotating angle of the unit. Instead, there is no phase change for co-polarized waves, no matter how the unit rotates. Liu et al. applied the PB phase to a tunable graphene focusing lens composed of the gold metasurface containing different rectangular apertures covered with monolayer graphene [76]. In order to realize the tunable focusing lens, the abrupt phase distribution of the lens needs to ensure that all transmitted light is in phase at the focal point, and as the Fermi level of graphene changes, the phase changes at different positions need to be different. For this lens, the total phase shift included the PB phase determined by the aperture rotation and the resonant phase determined by resonance. Therefore, apertures of different lengths were introduced because resonances from different lengths have diverse responses when changing the Fermi level of graphene. The lens’ focal length can change from 10.46 mm to 12.24 mm when the applied gate voltage increases from 0 V to 2 V.

Other patterned graphene structures, such as graphene nanoribbons and graphene split rings, have also been investigated to achieve phase modulation. Lu et al. proposed the monolayer graphene as nanoribbons, exploring a significant phase modulation of graphene [82]. The surface plasmon resonance of nanoribbon allows a significant phase shift, and both the Fermi level and the width of nanoribbon can affect the phase shift. However, the single nanoribbon cannot provide enough scattered amplitude for wave front control, as shown in Figure 5b. Researchers have attempted to combine the two nanoribbons as a unit cell to increase the intensity of the magnetic field. The unit cell’s normalized scattered wave amplitude is maintained at approximately 0.5. At the same time, the phase shift range of nearly 180° is sufficient for the modulation requirements, which can be used to build new ultra-thin graphene metasurface. Chen et al. arranged graphene split rings on a dielectric substrate to form tunable lenses for tunable polarization conversion and wave front control [80], as shown in Figure 5c. The collective oscillation of electrons was excited by incident light in graphene split rings, corresponding to efficient surface plasmons. Due to the structural properties of the graphene split rings, the mode of the surface plasmon transformed, resulting in a strong polarization conversion. For the efficient manipulation of the cross-polarized components, a linear variation of the phase shift (360°) along the metasurface must be achieved. The 360° phase modulation can be reached by adjusting the geometric parameters of split rings in 0.7–1.9 THz, while maintaining a high transmission amplitude. The overall modulation of the eight rings provides additional momentum for the cross-polarized transmitted light to deviate the wavefront from the normal metasurface. It is demonstrated that the deflection of the transmitted light decreases with increasing frequency, and the efficiency of the beam splitter can be dynamically tuned by tuning the graphene chemical potential.

In contrast to the transmissive modulator, a metallic mirror is added below the dielectric substrate in the reflector phase modulator. Therefore, the incident wave propagates in a multi-path between the top and the bottom layers, resulting in resonance on the top metasurface and the bottom reflector for a large phase modulation range. Kakenov et al. placed the monolayer graphene on the metal plate, which operates as a back reflector and gate electrode [77]. The gate voltage can adjust the surface plasma, altering the terahertz transmission in the dielectric layer and the excitation in the reflection surface, resulting in experimental phase modulation. Experiments show that a maximum of a 180° phase shift can be achieved at the resonant frequency of 1.2 THz. However, phase modulation at a single frequency cannot fully meet its application. Sherrott et al. combined graphene with a gold antenna array to increase the modulation bandwidth. The unit cells are arranged together with a small gap size to enhance the in-plane component of the electric field, increasing the sensitivity to graphene optical response. Therefore, by exploiting the tunable optical permittivity of graphene, this gate-tunable device is able to continuously shift the resonance peak from 8.24 μm to 8.81 μm. In the tuning range, the scattering phase is significantly modulated due to the effect of changing resonant states. The metasurface achieved dynamic phase modulation in the 8–9 μm band, gaining a modulation depth of more than 200° in the 8.5–8.75 μm band [74].

Sun et al. proposed another graphene coupling structure composed of multiple resonance modes to attain a more extensive depth, including dipole resonance, LC resonance, and SRR resonance [81], as shown in Figure 6a. These resonant modes are coupled with each other at a similar resonant wavelength, improving the sensitivity to the change of the surface plasmon; thus, the tunability of the composite structure is more robust than that of a single resonator with the same Fermi level variation range. The strong interactions between the composite coupling structure and the light enabled sufficient phase shifts at the resonant frequency range, achieving a maximum phase shift of 360° at 6.2 μm and a phase shift of no less than 270° at 5.7 μm to 6.1 μm.

Zhang et al. also realized a broadband phase modulator based on the graphene–metal hybrid structure [78]. The metal resonator is a disk with a gap in the middle, as shown in Figure 6b. It is well known that where the curvature of the conductor is large, the charge density becomes high. Therefore, the electric field is strongly localized at the edges of the disk. When the Fermi level increases, the carrier density in graphene increases simultaneously; the free carriers are driven by the electric field of the metal disk and gather at the edge of the graphene ribbon. As a result, the strongly enhanced electric field in graphene induces nonlinear effects, causing the phase shift to change quadratically with the electric field. Therefore, the phase of the reflected wave can range from −150° to 145° by tuning the Fermi level of graphene, and the maximum phase shift can approach 350°. Furthermore, a coding metasurface with a disk radius of 18 μm was proposed to modulate the far-field scattering of reflected waves. They defined the corresponding states of 0.05 eV and 0.4 eV Fermi levels as “0” and “1” codes, respectively. The phase difference between the two states was between 140° and −184.5° in the 4.15–4.65 THz band, which can realize the regulation of reflected light in the broadband range. For the Pancharatnam–Berry phase above, it is also applicable in reflective modulators. Ding et al. applied the PB phase for a reflector focusing lens by etching a rectangular aperture array on monolayer graphene [84]. The proposed lens comprised monolayer graphene, a metal reflector, and a dielectric layer. The continuous 2π phase shift and the high reflectivity enabled the lens to control the terahertz efficiently. The lens can achieve a high focusing efficiency of more than 60% and dynamically modify the focusing intensity and the focal length.

Independent of resonant structure, Chen et al. designed a graphene metasurface based on the tunable Brewster angle with a high modulation depth and a large bandwidth [79], as shown in Figure 6c. It demonstrated that p-polarized light experiences zero reflection when incident at the Brewster angle, and the reflected light undergoes a 180° phase shift when the incident angle crosses the Brewster angle. Due to the complex conductivity of graphene, the reflection amplitude at Brewster’s angle is not zero, and the phase changes gradually as the incidence angle approaches Brewster’s angle, which is slightly different from the regular Brewster’s law. Moreover, they also experimentally applied voltage to the graphene monolayer, demonstrating that the Brewster angle of the metasurface could be tuned from 65 to 71° by varying the conductivity of graphene. When the incident angle is fixed, the maximum phase modulation of approximately 140 ° can be achieved in 0.5–1.6 THz by using the phase jump characteristics of Brewster angle.

### 3.3. Terahertz Photodetectors

Owing to its high-quality mechanical flexibility, good thermal conductivity, and ultra-high carrier mobility, graphene has excellent potential for wideband photodetection in the terahertz band. The core principle of the photodetector is to convert the incoming light signal into an electrical signal, which means that the conductivity of the material changes during the process. Several different mechanisms by which this can be accomplished in graphene have been reported, mainly including the photovoltaic effect (PVE) [85], the photo-thermoelectric effect (PTE) [86,87], the bolometric effect (BE) [88,89], and the plasma-wave-assisted mechanism in graphene FET [90,91,92,93]. In order to improve the performance of detectors, most current research focuses on improving the absorption of electromagnetic waves by materials, reducing dark current, reducing response time, and broadening the working bandwidth as much as possible. We summarize several graphene-based methods to improve detection performance and to introduce the latest progress accordingly.

It is worth noting that graphene shows high wave locality, and it supports the propagation and the dynamic tunability of surface waves. Therefore, graphene can be regarded as a very encouraging platform for integrated optoelectronics of terahertz antennas, such as resonant antennas [94,95], hybrid plasmonic antennas [96,97,98], and leaky-wave and reflect array antennas [99,100]. Combining single-layer or multilayer graphene antennas with photodetectors surpasses conventional detectors. In the past, integrated graphene antennas with various geometries have been investigated for detection purposes [101,102,103]. Additionally, employing graphene in an antenna integrated photodetector can improve signal coupling and other characteristics of the photodetector. Castilla et al. proposed a novel THz radiation detector using a dual-gated dipole antenna with a gap of ∼100 nm [104]. As shown in Figure 7a, the incoming radiation is intensely concentrated at the PN junction of the graphene channel above the antenna. This highly sensitive detector has an equivalent noise power of $80\;\mathrm{pW}/\sqrt{Hz}$, a response time of less than 30 ns, and a wide operating bandwidth (1.8–4.2 THz) at room temperature. Figure 7b,c show the specific performance of the device. Even more remarkable is that it achieves a combination of high performance that is not currently possible with advanced detectors. Tunnel field-effect transistors based on bilayer graphene (BLG) can also be used for sensitive THz detection [105]. They coupled the antenna with a tunnel junction created by the electrically tunable properties of BLG (Figure 7d), enabling low-noise ($0.2\;\mathrm{pW}/\sqrt{Hz}$) and high responsivity (>4 kV/W) detection, as shown in Figure 7e,f. A room temperature THz detector with a wide bandwidth (~100 GHz), high speed (several hundred ps), and high sensitivity (equivalent noise power ≤ $120\;\mathrm{pW}/\sqrt{Hz}$) at 3.4 THz were also achieved by using the antenna-coupled graphene FET [106]. Figure 7g–i shows how it works and its excellent detection performance. These excellent features open unique perspectives for high-speed applications in many research fields such as ultrafast nano-spectroscopy, high-speed communications, quantum science, and coherent control of quantum systems.

In addition to the coupling efficiency mentioned above, detectors made of single materials tend to have powerful dark currents, resulting in a large NEP and a low detection rate. Graphene combines with other 2D materials to form an interlayer van der Waals heterostructure. The high-energy potential barrier formed can effectively inhibit the disorderly motion of photoelectrons to reduce dark noise. Related work has made remarkable progress across a broader spectrum. Xu et al. proposed a photodetector with a few-layer MoS_2_/glass-to-graphene heterostructure synthesized by a layer-by-layer transfer technique [107]. The ohmic contact formed between MoS_2_ and graphene significantly improved the charge transfer efficiency in the device. Meanwhile, the results showed that the band arrangement of layered heterostructures can be controlled by lattice engineering of 2D nanosheets to improve the photoelectric properties. Similarly, Guan et al. showed that heterojunction formation by doping graphene could improve the light-detection capability of detections in their device based on ferroelectric polarization of $L_{i}NbO_{3}$ [108]. The detectors’ performance, such as detectivity and responsivity, improved by several orders of magnitude compared to previously reported devices. Although advances have focused on visible, near-infrared, and mid-infrared wavelengths, it is exciting that theoretical calculations have already shown that graphene can be extended to THz waves by altering the band gap of graphene [109,110]. Caoet al. achieved a broad photo response from VIS (532 nm) up to the THz region (2.52 THz) in the reduced graphene oxide (RGO) and Si nanowire (SiNW) array heterojunction photodetector [111].

Other methods, such as defect engineering [112], improvement of substrate materials, and introduction of quantum dot structures in graphene field-effect transistors [113,114,115], can effectively improve the detection performance of graphene detectors. Graphene is not the only two-dimensional material that efficient terahertz photodetectors rely on. Others such as black phosphorus (BP), bismuth and telluride also perform well in detectors. However, due to its broad spectral properties (UV to THz), graphene’s potential applications in photodetectors cover the mid-infrared and the terahertz bands, making it attractive to researchers in many different fields. Although the performance improvement methods mentioned in this section can enhance the performance of THz detectors by several orders of magnitude, the intrinsic properties of the materials themselves are still the fundamental factors. Table 3 presents the figure-of-merits of photodetectors based on graphene and some representative 2D materials. It can be seen that graphene devices perform well even in broadband bands, especially in terms of the response time, which is still improved for wider commercial applications. Thanks to these excellent detection properties, graphene-based sensors are widely used to detect material properties and for biomedical and chemical detection [116,117,118,119,120].

In addition to improving the performance parameters of the detector itself, an urgent challenge is how to apply it to the existing communication systems and achieve multi-platform compatibility without affecting the excellent performance of devices. Asgari et al. proposed an efficient optical detection method for monocrystalline and polycrystalline graphite, which was based on chemical vapor deposition (CVD) growth, with equivalent noise power < 1 $\mathrm{nW}/\sqrt{Hz}$ and response time ~5 ns at room temperature [121]. Tuning in the entire THz range (0.1–10 THz) can be achieved by selecting the resonance of an on-chip patterned nanoantenna. CVD helps to relax geometric and technical constraints at terahertz frequencies, while retaining chip-level scalable CMOS technology compatibility. Other examples include a high-responsivity graphene photodetector integrated with a silicon microring resonator [131]. Under critical coupling, 90% light absorption was achieved in the SLG channel of ~6μm along the Si waveguide, with a voltage responsivity of ~90 V/W. At the same time, the receiver sensitivity guaranteed a bit-error-rate of only $10^{-9}$. By introducing an ultra-thin and wide silicon-graphene hybrid plasmonic waveguide [132], a high-performance waveguide photodetector based on the radiant heat/photoconductivity effect was realized. When operating at 2 μm, the photodetector’s responsivity was ~70 mA/W and the 3 dB bandwidth was >20 GHz. When operating at 1.55 μm, the bandwidth of the photodetector was 40 GHz and the photodetector’s responsivity was ~0.4 A/W even with a low bias voltage of −0.3 V. All these measures pave the way for the future development of graphene-integrated devices.

### 3.4. THz Generation

Compared with the graphene THz devices mentioned above, graphene-based THz generation is still in a much earlier stage. Because the traditional THz radiation equipment is complicated and expensive, the generation of THz is significantly more challenging than generating more extended wavelength sources (microwaves). Table 4 shows the recent evolution of THz generation, and we could find that different structures combined with different mechanisms result in different characteristics and performances of THz generators. In the following content, we briefly discuss the recent progress of THz generation based on different mechanisms, including the generation and the tunability of THz sources based on different mechanisms.

High harmonic generation could convert the incident THz signals into signals with a much higher frequency, which is critical for generating THz sources with a specific frequency [141]. Figure 8a shows the schematic of the odd-order harmonics emitted from the monolayer graphene [26]. The transmission rate of the fundamental wave is more than 90% for the generated third, fifth, and seventh harmonics; the transmission rate is slightly lower than 10^−1^, 10^−2^, and 10^−3^, respectively, in the normalized spectral amplitude for 0–2.5 THz (Figure 8b). As the pump power increases, the HHG efficiency tends to saturate, caused by the dissipative nature of the interaction between the THz field and the carriers (Figure 8c).

Besides, the improvement of HHG efficiency is also significant. Because of the graphene tunability [142,143,144], the HHG efficiency could be improved by tuning the Fermi level of graphene. For example, Kovalev et al. designed a gated graphene device for tuning the high harmonic generation [26]; graphene was grown by CVD with a single layer and placed on a 1 mm thick substrate. Two metal electrodes were connected at both sides of the graphene as source and drain, and the gating was controlled by another pair of electrodes, as shown in Figure 8a. In the experiment, there was a multicycle terahertz field with a central frequency of 0.3 THz incidents on the sample, the peak electric field was as high as 80 kV/cm. In Figure 8b, the frequency-domain spectra of the THz field that transmitted through the graphene was detected. The spectra contains odd-order harmonics of the fundamental frequency (0.3 THz) up to the seventh order; the HHG efficiency could be tuned by the Fermi level of graphene, as shown in Figure 8b. By tuning the Fermi level of graphene and achieving a THG efficiency enhancement of approximately two orders by using electrical gating of graphene (Figure 8c), the resonant effects of metamaterials induced by their geometry could enhance light–matter interactions, including hole arrays [145,146], gratings [27,147], and split-ring resonators [148,149]. Thus, the combination of metamaterial and graphene could significantly improve the HHG efficiency of graphene. Furthermore, there are many studies for HHG enhancement with graphene metasurfaces [141,150]. For example, Guo et al. designed a graphene-covered groove grating structure to generate third-order harmonics [151]. The grating groove structure localized and strongly enhanced the electric field in the groove, which improved the nonlinear optical response. Compared with the ordinary flat substrate covered by graphene, the generated third harmonic power increased by nearly 20 times. The conversion rate of the third harmonics increased with the enhancement of the incident THz field. Later, Deinert et al. reported a graphene–metal grating platform to provide field enhancement experimentally. Compared with the device without metal grating, the nonlinear efficiency of the graphene–metal grating structure improved by 50 times and by three orders of magnitude for the third harmonic [152].

Compared with HHG, the THz generation based on the graphene DFG does not need the incident of other THz fields. It is achieved using laser sources at near-infrared frequencies, with the energy conversion from the input optical energy to the THz signal [153]. In 2014, Yao et al. proposed a scheme to generate THz plasmons with DFG. The theoretical result showed that the efficient generation of THz plasmon could be achieved through the DFG process of graphene, and the generated THz plasmon could be changed by adjusting the incident angle and the graphene Fermi level. In addition to the case of the free-space pump, they also discussed the composite structure of graphene waveguides, demonstrating the feasibility of this method for on-chip generation of THz plasmons [53]. Later, Constant et al. proved experimentally the all-optical generation of THz plasmons. Two visible light pulses and a generated THz plasmon matched both the wavevector and the energy. The result showed that the excited THz plasmon could have a defined wavevector and direction across an extensive frequency range, with a conversion efficiency of 10^−5^ [135].

However, free-space pump-based THz generation is not suitable for integration. In 2013, Sun et al. proposed a method to generate THz plasmons with the graphene composite waveguide. Unlike the THz plasmon generation by DFG, the FWM as a third-order nonlinear excitation of the THz plasmon was presented in their work. The numerical calculation showed that the generated THz plasmon power reached 67 W when the input pump power was 1 kW. Nevertheless, other factors in the numerical calculation were not considered, such as the low third-order nonlinear conversion efficiency [134], other FWM processes, or the graphene Fermi level, so the actual conversion efficiency could be much lower. Until recently, experimental results for graphene FWM based THz plasmon had not been reported. However, in 2018, both theoretical and experimental results for THz plasmon generation on a graphene composite waveguide were reported: Chen et al. designed a novel graphene/AlGaAs SPW structure for nonlinear generation of THz plasmon and found that phase-matching terahertz wave frequency varied from 4 to 7 THz when the Fermi level changed from 0.848 to 2.456 eV [136]. Later, Yao et al., reported THz plasmon generation and active manipulation on the chip-scale integrated waveguide. As shown in Figure 8d, bilayer graphene was deposited on a silicon nitride waveguide and a layer of Al_2_O_3_ as a thin dielectric barrier. The result showed that when the gate voltage increased from −0.7 V to −0.3 V, the DFG signal blueshifted from 1607.2 to 1601.3 nm, with the intensity increasing from 0.28 to 0.37 a.u. (Figure 8e), and its tunability was only limited by the pump amplifier optical bandwidth. Besides, Figure 8f shows that when the V_G_ is 0 V, the measured spectra of the 1593.2 nm increased linearly from 0 to 0.3 a.u., with the pump power rising from 0 to 32 mW. With the assistance of broadband tunable THz plasmon, the on-chip THz plasmon could have strong potential for miniaturized hybrid telecom–THz circuits [137].

In addition to the above methods of generating THz by graphene nonlinearities, THz emission could also be generated directly. In 2019, Andersen et al. reported the tunable generation of THz vibrations by electrically gated graphene devices [138]. Later, Li et al. reported THz light emission based on current-driven graphene plasmonic oscillations. The device consisted of nanoribbons with widths of 530 and 810 nm, arranged in an array of approximately 50% duty cycle graphene grown by CVD, and transferred to the oxidized Si substrates; graphene on the substrate was etched into the desired nanoribbon pattern by electron beam lithography and reactive ions and then the metal contacts was fabricated, as shown in Figure 8g. The plasmon resonance was excited by the injected current and then radiated into the far field. In Figure 8h, they tested the devices with widths of 530 and 810 nm; the result showed that the narrower nanoribbon device could consistently emit at higher frequencies with the same carrier density, and the frequency could also be tuned by the gate voltages. It indicated that the THz emission efficiency could be tuned by the structure of graphene nanoribbons and the carrier density (tuned by fermi level) [139]. Besides, Figure 8i shows the normalized emission peaks with different electrical power ($P_{in}=\left(V_{DS}^{2}\right)/R$), and the emission initially increases with the input power, but when the input power increases to 390 mW, the plasmonic emission gradually decreases. Recently, Justin et al., proposed a graphene-based plasmonic nano-generator, which was based on a gated high electron mobility transistor. The THz plasmon was generated on the gate when the asymmetric boundary conditions with source voltage and drain current were satisfied. The amplitude and the frequency modulation could be modulated with the bandwidths theoretically approaching 200 GHz [140].

As mentioned above, graphene-based THz generation mainly relies on the nonlinear properties of graphene, including THG, HHG, and DFG. Due to the thickness of a single atomic layer of graphene, an input pump with high power is required to excite graphene nonlinearity. Thus, a pulse pump with high peak power is generally used for THz generation. Obviously, the output THz is mainly in the form of a pulse signal. Thus, in order to further develop graphene-based THz generators and optimize their performance, it is very important to learn from other pulse generation and optimization methods [154,155,156]. Furthermore, these methods will provide important references for the development of graphene-based THz pulse generation.

## 4. Conclusions and Future Perspective

This review has summarized the fundamental aspects of graphene in terms of linear and nonlinear properties and the optical modulation mechanism of graphene metamaterial. Furthermore, it has reviewed the applications of graphene metamaterial in the THz region, including terahertz amplitude and phase modulators and photodetectors and terahertz generation. The metamaterial structure provides an effective way to enhance light–graphene interactions in a tunable and a feasible fashion, with the advantages of high efficiency, integration, multi-functionality, and low energy consumption. However, limited by the complex graphene fabrication technology and the incompatibility with a wide range of substrates, there are still challenges for the practical utilization of graphene terahertz devices.

Advanced research on graphene metamaterials in terahertz is developing toward intelligence and integration, and we further note some potential directions. In addition to graphene, other new materials have also been extensively investigated for THz devices, such as 2D transition metal dichalcogenides (TMDCs) [157], BP [158], and topological insulators. These new materials provide new light–matter interactions and mechanism platforms by combining with metamaterial structures, leading to an integrated THz meta-device, and even new THz phenomena. In structural design, artificial intelligence and machine learning are expected to reshape the smart design of meta-devices. Many recent studies have been devoted to metastructure design methods and emerging material platforms, demonstrating ANN-assisted predictions for topological transitions and an inverse design of 2D-topological insulators [159,160]. Compact and highly tunable THz devices are vital for developing miniaturized and integrated THz technology. Chip–scale integrated devices for terahertz generation and phase modulation were recently demonstrated [137,161]. They are all based on the waveguide, which could be easily integrated into chips of THz electronics. The combination of two-dimensional materials or 2-DEG and waveguides provides much potential for THz chip–scale integrated devices. It is expected to achieve fully integrated on-chip THz devices, which will significantly promote the miniaturization of THz devices and the development of THz communication devices.

## Figures and Tables

**Figure 1 nanomaterials-12-02097-f001:**
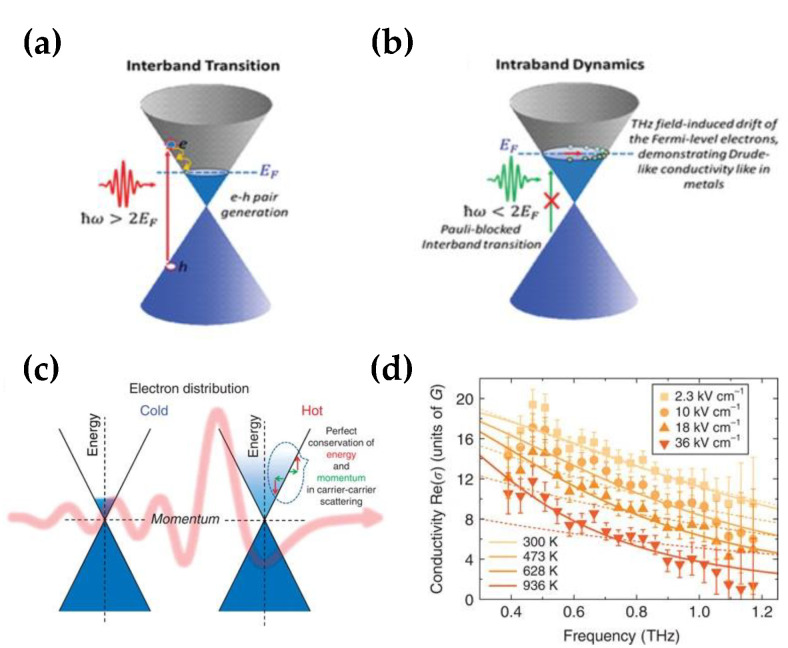
Interaction of graphene with the incident field. (**a**) Interband transition when the photon energy (ℏw) is more than twice the Fermi level (*E_F_*); (**b**) Intraband dynamics when the incident energy (ℏw) is lower than twice the Fermi level (*E_F_*); (Reprinted with permission from ref. [15]. Copyright 2020, Advanced Optical Materials) (**c**) Cold− and hot−carrier distribution in graphene, before and after incident in the THz field; (**d**) Conductivity for different carriers’ temperatures and different THz frequencies. (Reprinted with permission from ref. [50]. Copyright 2015, Nature Publishing Group).

**Figure 2 nanomaterials-12-02097-f002:**
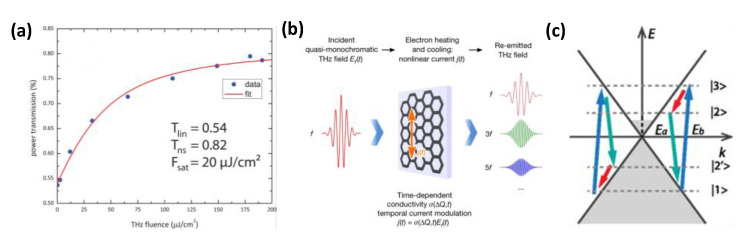
THz nonlinearities in graphene. (**a**) Nonlinear saturable absorption for THz; (Reprinted with permission from ref. [52]. Copyright 2013, American Chemical Society) (**b**) High harmonic emitted from the monolayer graphene excited by a multicycle THz wave [25]; (**c**) Schematic of DFG process. *E_a_*: photon energy of signal, *E_b_*: photon energy of pump. (Reprinted with permission from ref. [53]. Copyright 2014, American Physical Society).

**Figure 3 nanomaterials-12-02097-f003:**
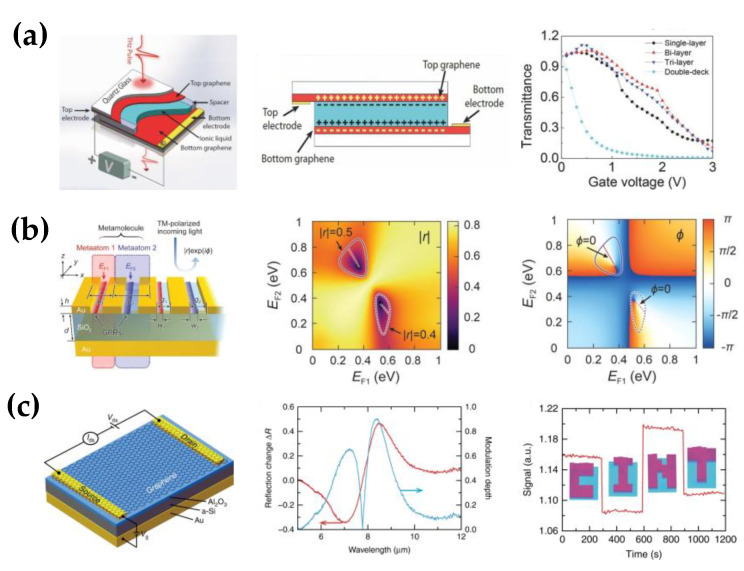
Electronically controlled terahertz amplitude modulator based on graphene. (**a**) Sandwich structure of two-layer graphene and ionic liquid, holes, and electrons accumulate at the ionic-liquid/graphene interfaces when a voltage is applied, the transmission modulation from structures with different numbers of the graphene layer; (Reprinted with permission from ref. [63]. Copyright 2015, Wiley-VCH) (**b**) Plasmonic metamolecule composed of independently controlled graphene meta-atoms, the graphene Fermi levels applied to meta-atoms are defined by E_F1_ and E_F2_, a great amplitude modulation with the fixed phase of $0^{{}^{\circ}}$ and a 360° phase shift with the fixed amplitude of 0.5 can be realized by adjusting the E_F1_ and E_F2_ Reprinted with permission from ref. [67]. Copyright 2020, American Chemical Society; (**c**) Hybrid graphene metasurface for high-speed amplitude modulation where a-Si serves as part of the back gate electrode, the variation of reflection and modulation depth for V_g_ = 7 V and −3 V under different incident wavelengths, and spatial reflection patterns of “CINT” at λ = 8.3 μm by selectively applying different gate voltages. (Reprinted with permission from ref. [66]. Copyright 2015, Nature Publishing Group).

**Figure 4 nanomaterials-12-02097-f004:**
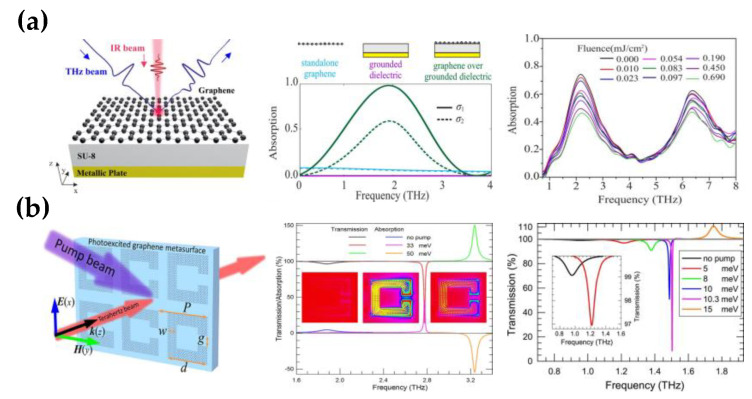
Optically pumped terahertz amplitude modulator based on graphene. (**a**) The metasurface is composed of graphene deposited on SU-8 substrates, the absorption spectra of three structures for perfect absorption conditions graphene conductivity ($\sigma_{1}$) and for another case ($\sigma_{2}$) that does not fulfill above conditions, the photoexcited spectra of a graphene-based structure under fluence from 0 to 0.690 mJ/cm^2^; (Reprinted with permission from ref. [51] Copyright 2019, American Chemical Society) (**b**) The periodic array of split-ring resonators made of photoinduced monolayer graphene, the magnetic resonances of graphene SRRs under optical pumping, transmission spectra for unpumped and pumped graphene with different Fermi levels. (Reprinted with permission from ref. [65] Copyright 2018, American Chemical Society).

**Figure 5 nanomaterials-12-02097-f005:**
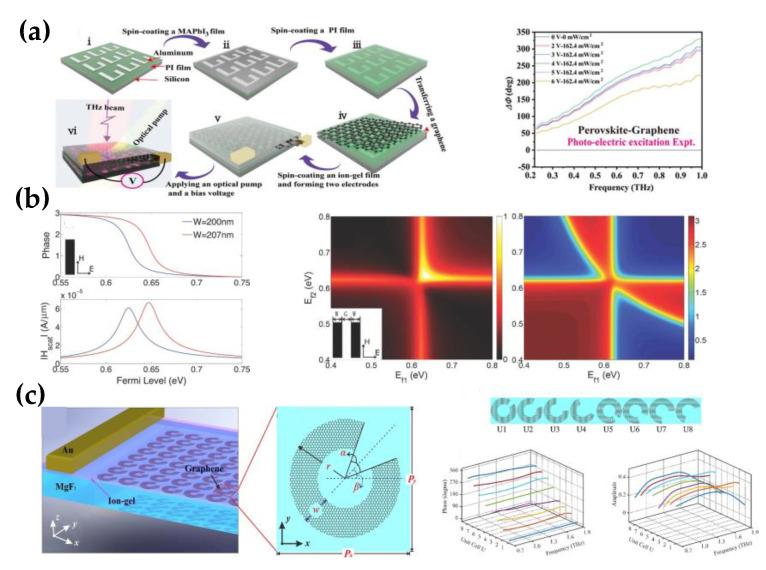
The transmissive tunable graphene terahertz phase modulator. (**a**) Fabrication process of the photoelectric metasurface composed of graphene, polyimide, and perovskites, transmission phase difference of the perovskite−graphene metadevice under photo−electric excitation; (Reprinted with permission from ref. [69] Copyright 2022, American Chemical Society) (**b**) Phase shift and magnetic field amplitude of scatters by graphene plasmonic resonance for an infinitely long nanoribbon versus the Fermi level, where the magnetic field amplitude drops sharply from the resonance region; the magnetic field amplitude and phase shift are enhanced under two independent Fermi levels of two parallel nanoribbon; (Reprinted with permission from ref. [82] Copyright 2014, Wiley−VCH) (**c**) Monolayer graphene split-rings are transferred on the MgF2 with a layer of ion-gel on top for polarization modulation, the vertical view of split-ring with geometrical parameters; the eight units with different opening angles form one supercell, covering 360° phase response with an amplitude of nearly 0.4. (Reprinted with permission from ref. [80] Copyright 2019, Elsevier).

**Figure 6 nanomaterials-12-02097-f006:**
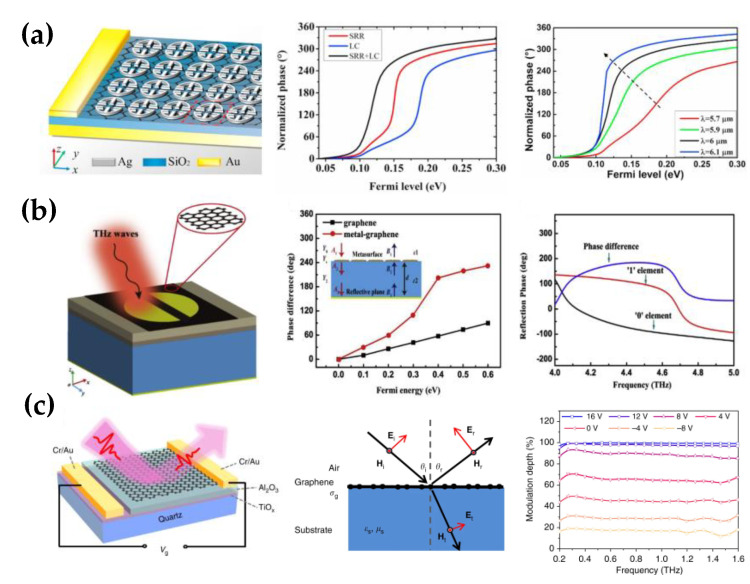
The reflective tunable graphene terahertz phase modulator. (**a**) Graphene−metal hybrid modulator was composed of the silver split ring and nanorod with monolayer graphene as active material, the phase tunability of SRR and LC is smaller than that of the coupling structure; the phase modulation range and smoothness under four incident wavelengths, the maximum shift of nearly 360° can be achieved at 6.1 μm; (Reprinted with permission from ref. [81] Copyright 2021, Elsevier) (**b**) Reflection unit consisted of a graphene sheet and a gold disk separated in the middle, comparison of phase difference for the bare graphene and the metal−graphene metasurface at 4.5 THz; the reflection phase distribution of ‘0’ (*E_F_* = 0.05 eV) and ‘1’ (*E_F_* = 0.4 eV) elements in coding metasurface, the phase difference was approximately 180° at frequency from 4 THz to 5 THz; (Reprinted with permission from ref. [78] Copyright 2021, Elsevier) (**c**) Tunable Brewster reflection phase modulator based on graphene/quartz surface with broad bandwidth and high modulation speed; diagram of the incident light from air to graphene/silica substrate, which was the operation principle for the Brewster modulator; the broadband phase modulation can be achieved under different gate voltages in the frequency range of 0.2−1.6 THz. (Reprinted with permission from ref. [79]. Copyright 2018, Nature Publishing Group).

**Figure 7 nanomaterials-12-02097-f007:**
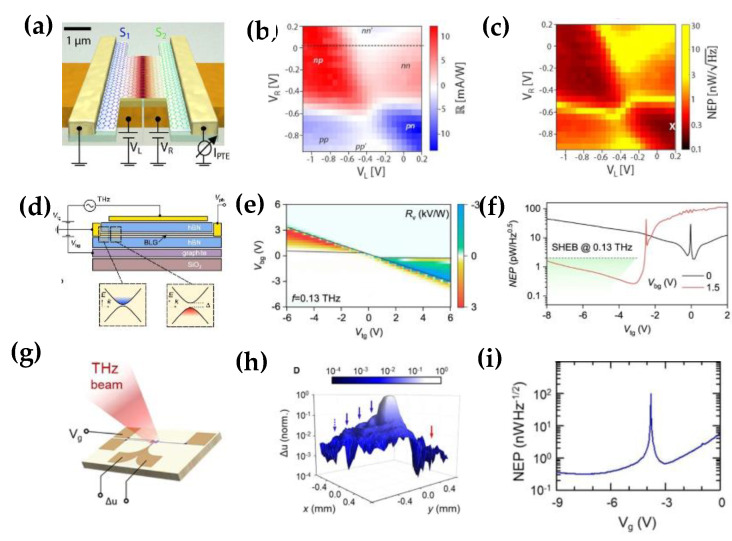
(**a**) Schematic diagram of the antenna−integrated pn junction device; (**b**) Relation between the photo response and the applied voltage of the two branches of the antenna(VL and VR) when the incident frequency is 2.52 THz; (**c**) Noise-equivalent power (NEP); (Reprinted with permission from ref. [104] Copyright 2019, American Chemical Society) (**d**) Schematic diagram of dual-gated BLG transistor in hBN package; (**e**) Relationship between photo response and gated voltage (V_bg_ and V_tg_) when the incident frequency is 0.13 THz; (**f**) Noise−equivalent power (NEP) at given V_bg_; (Reprinted with permission from ref. [105]. Copyright 2021, Nature Publishing Group) (**g**) Schematic diagram of antenna coupled GEFT; (**h**) Photo response represented by Δu; (**i**) Noise−equivalent power (NEP) calculated as a function of V_g_. (Reprinted with permission from ref. [106]. Copyright 2021, Nanophotonics).

**Figure 8 nanomaterials-12-02097-f008:**
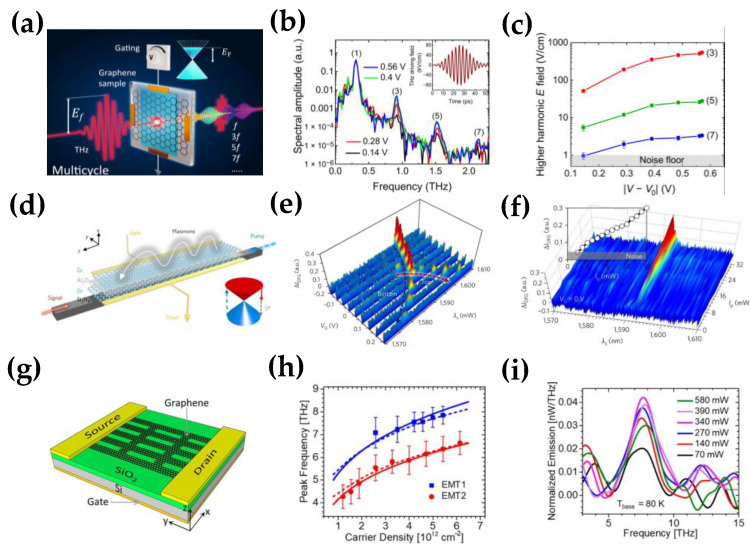
THz generation. (**a**) Gated graphene sample device for tuning the high harmonic generation. (**b**) Amplitude spectra of terahertz wave and high harmonics transmitted through the graphene. (**c**) Tunability of HHG efficiency as a function of gating voltage, Inset: the driving terahertz signal; (Reprinted with permission from ref. [26] Copyright 2021, American Association for the Advancement of Science) (**d**) Graphene on silicon nitride waveguide (GSiNW) architecture, Inset: Dirac cone structure of the DFG process. (**e**) Tuning of the graphene terahertz plasmon signal. (**f**) DFG signal intensity increases linearly with the pump rising; (Reprinted with permission from ref. [137]. Copyright 2018, Nature Publishing Group) (**g**) Graphene-nanoribbon structure of THz source. (**h**) Tunability of plasmonic emission of the device. (**i**) Pump power dependence of plasmonic emission from graphene nanoribbons. (Reprinted with permission from ref. [139] Copyright 2019, American Chemical Society).

**Table 1 nanomaterials-12-02097-t001:** Summary of THz amplitude modulators.

Work	Frequency	Mod. Speed	Depth Mod.	Ref.
Lee et al.	0.1–0.6 THz	100 kHz	47%	2012 [58]
Valmorra et al.	0.8–1.75 THz	-	11.5%	2013 [59]
Degl’Innocenti et al.	2.2–3.1 THz	-	18%	2014 [60]
Gao et al.	0.38–0.5 THz	-	50%	2014 [61]
Liu et al.	3.5–5.5 THz	40 MHz	60%	2015 [62]
Wu et al.	0.1–2.5 THz	-	93%	2015 [63]
Jung et al.	0.4–0.8 THz	-	49.3%	2018 [64]
Fan et al.	1.5 THz	-	92%	2018 [65]
Zeng et al.	34.09–36.59 THz	1 GHz	90%	2018 [66]
Han et al.	42.86 THz	-	80%	2020 [67]
Tasolamprou et al.	2.0–2.6 THz	2.79 ps	40%	2019 [51]
Choi et al.	0.75–1.05 THz	1.83 ps	80%	2021 [68]
Yang et al.	1.66–1.74 THz	-	70%	2022 [69]
Yao et al.	1.6–1.75 THz	-	77%	2021 [70]

**Table 2 nanomaterials-12-02097-t002:** Summary of THz phase modulators.

Work	Frequency	Mod. Loss	Phase Mod.	Ref.
Lee et al.	0.5–0.75 THz	-	32.2°	2012 [58]
Gao et al.	0.1–0.8 THz	18 dB	57°	2014 [61]
Balci et al.	11.82 GHz	60 dB	90°	2018 [72]
Dabidian et al.	39 THz	10 dB	55°	2016 [73]
Sherrott et al.	33.3–37.5 THz	-	237°	2017 [74]
Park et al.	50.42 THz	37 dB	180°	2017 [75]
Liu et al.	0.75 THz	41.62 dB	91.8°	2018 [76]
Kakenov et al.	1.2 THz	50 dB	180°	2018 [77]
Zhang et al.	4.15–4.65 THz	15.5 dB	184.5°	2018 [78]
Chen et al.	0.5–1.6 THz	12 dB	140°	2018 [79]
Chen et al.	0.7–1.9 THz	8 dB	360°	2019 [80]
Yang et al.	0.2–1.0 THz	-	346°	2021 [69]
Sun et al.	5.7–6.1 um	-	330°	2021 [81]

**Table 3 nanomaterials-12-02097-t003:** Summary of THz photodetectors.

Detector Type	Frequency	NEP	Response Time	Ref.
Graphene	1.8–4.2 THz	$80\;\mathrm{pW}/\sqrt{Hz}$	<30 ns	2019 [104]
Graphene	0.13 THz	$0.2\;\mathrm{pW}/\sqrt{Hz}$	-	2021 [105]
Graphene	3.4 THz	$\leq 120\;\mathrm{pW}/\sqrt{Hz}$	≈10^−1^ ns	2021 [106]
Graphene	2.8 THz	$<103\;\mathrm{pW}/\sqrt{Hz}$	5 ns	2021 [121]
Graphene	0.12 THz	$100\;\mathrm{pW}/\sqrt{Hz}$	-	2018 [122]
Graphene	0.1–0.4 THz	$34\;\mathrm{pW}/\sqrt{Hz}$	-	2017 [123]
Graphene	2 THz	$1700\;\mathrm{pW}/\sqrt{Hz}$	-	2015 [124]
Graphene	0.33 THz	$51\;\mathrm{pW}/\sqrt{Hz}$	-	2017 [125]
BP	3.4 THz	$7000\;\mathrm{pW}/\sqrt{Hz}$	-	2019 [126]
BP	0.29 THz	$138\;\mathrm{pW}/\sqrt{Hz}$	800 ns	2020 [127]
PtTe2	0.12 THz	$<10\;\mathrm{pW}/\sqrt{Hz}$	≈1.7 × 10^4^ ns	2019 [128]
PtTe2	0.3 THz	$57\;\mathrm{pW}/\sqrt{Hz}$	≈10^3^ ns	2020 [129]
Bi2Se3	0.3 THz	$0.36\;\mathrm{pW}/\sqrt{Hz}$	<6 × 10^4^ ns	2018 [130]

**Table 4 nanomaterials-12-02097-t004:** Summary of THz generation.

Work	Structure	Mechanism	Efficiency	Ref.
Hafez et al.	SiO_2_ substrate	HHG	10^−3^	2018 [25]
Theodosi et al.	metasurface	THG	10^−2^	2022 [133]
Kovalev et al.	gate-tunable graphene	HHG	~10^−2^	2021 [26]
Sun et al.	waveguide	FWM	6.7%	2014 [134]
Constant et al.	SiO_2_ substrate	DFG	6 × 10^−6^	2016 [135]
Chen et al.	graphene/AlGaAs waveguide	DFG	4 × 10^−6^	2018 [136]
Yao et al.	gate-tunable graphene waveguide	DFG	~0.6 × 10^−4^	2018 [137]
Andersen et al.	gate-tunable graphene	current-driven	-	2019 [138]
Li et al.	gate-tunable graphene nanoribbons	current-driven	~2 × 10^−8^	2019 [139]
Justin et al.	gate-tunable graphene	current-driven	-	2021 [140]

## Data Availability

Not applicable.

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
