# Peer review of "Dynamic and Active THz Graphene Metamaterial Devices"

_nanomaterials, 2022, doi:10.3390/nano12122097_

Round 1

Reviewer 1 Report

The topic has been receiving a lot of attention in recent years, and has already been published in other journals.The submitted manuscript is a short review dealing with amplitude/phase modulators and signal generation/detection devices of various structures that can be used, especially in the THz band, using the unique and superior optical and electrical properties of graphene.

Specifically, the description focused on the development trend and direction of the proposed devices based on the combination of graphene and metamaterials usable in the range of THz frequency. Although this manuscript is not very specific or detailed about the theoretical model that describes the optical properties of graphene, it is neatly described without any fuss to provide an understanding of the physical properties of graphene.

In addition, the development status of amplitude/phase modulators and THz wave detector/sources using THz graphene-based metamaterials with such optical linear and non-linear properties were introduced.

When explaining the modulation devices, the physical properties related to the key operating principle and the performance characteristics of the device were described, and the results of various structures that can compensate for the physical limitations of graphene were also introduced to have an overall systematic configuration.

However, compared to the content of the amplitude and phase modulators, the introduction of the THz wave detectors and generators seems insufficient. Therefore, figure-of-merits (FOM) describing the performance of the detectors and sources and the tables in which the reported performances are summarized are necessary to improve the completeness of the manuscript.

In addition, since the heterojunction structure-based photodetectors (are shown in Fig. 9) are not related to the detection of the THz wave detectors, it is likely to be helpful to maintain the consistency of the topic by replacing the content with related experiments and calculation results.

*Also, like ref.1(in page 18 of 25), it is necessary to review minor typos in the manuscript.

Reviewer 2 Report

The authors presented a review on the THz applications of graphene and graphene-based metamaterials with a main focus on optically and electrically actuated THz amplitude and phase modulations, high harmonic generation and photodetection. I find the topic interesting, especially with respect to graphene metamaterial applications and the review presents a lot of information on the state-of-the art graphene-based THz devices.

However, there are some critical points, which prevent me to recommend this manuscript for publication in its present form: 1) The English is not good and has to be improved by proof reading of a native English speaker; 2) I would recommend the authors (may be already in the introduction) to make a comparison of the addressed THz performance of graphene metamaterials with other materials used for THz applications. Is graphene the only alternative for THz devices? What is known for other materials? This will allow the reader for a better orientation in the “THz-World” and for ranking between different THz materials; 3) In part 2 the authors present and discuss basic linear and nonlinear optical properties of graphene, providing some formulas (1-3, 5) to calculate the dynamic (Eq. 1) and intraband (2) optical conductivity as well as nonlinear absorption (3) and second-order susceptibility (5). Of course, they have cited the corresponding papers, but, I guess, some qualitative explanations and steps to obtain and understand these formulas would help the reader; 4) Concerning the figures: they contain a lot of information and in many cases the authors do not address the parts of figures. Moreover, the figure legends also do not describe the parts of figures, e.g., Fig. 5. In text the authors mentioned Fig. 5a) and Fig. 5b), which both are comprised of 3 sub-figures. In the context sometimes one gets a “punch-line” by looking on these figures, but sometimes one loose it. Note, that not all figures are self-explaining and one needs a better description of all figures, including sub-figures. May be a reduction of the number of sub-figures could be a solution in some cases; 5) In general, the manuscript needs better explanation of the underlying physics behind of applications, otherwise the review reduces simply to a kind of “handbook”, in which only the numbers and references are sufficient.  

Summarizing, I believe the manuscript can be published in Nanomaterials (MDPI) after authors considered my comments and changed the manuscript accordingly.

Reviewer 3 Report

This is an interesting work where THz graphene is explored for metamaterials. Some of my recommendations are:

In the introduction, some brief investigation is needed to be discussed based on THz transmission lines specifically NLTL and MSTL. recently it was well established that some nonlinear transmission lines also perform very well in this regard.

Section 2 need to be reduced considerably and the equations need to be modified in a compact manner.

some overview needed to be placed on graphen based THz antennas as well.

In case of THz generation, which is a type of pulse generation some state of the art pulse generators need to be placed as well. You need to discuss all type of gaussian pulses. you can follow the work in:  "Theoretical and experimental analysis of pulse compression capability in non-linear magnetic transmission line" and like this for corresponding pulse generation capabilities.

Material exploration is one of the interesting topic in this regard as well. Need to be discussed briefly as well.

Round 2

Reviewer 1 Report

 The authors have adapted the manuscript content appropriately by accepting the reviewers' comments, and the added explanations and tables also seem to help improve the understanding of the topic of the manuscript. Therefore, in my opinion, the manuscript deserves publication in nanomaterials.

Author Response

No further comments from reviewers

Reviewer 3 Report

All my comments are well addressed.

Thanks for your detailed modifications.

Author Response

No further comments from reviewer